

TITLE
Improving measurements of microbial growth, death, and turnover by accounting for extracellular DNA
in soils
Keywords: soil carbon cycling, microbial death, soil microbial processes, microbial temperature response,
microbial growth optimum
AUTHORS
Jörg Schnecker[1], Theresa Böckle[1], Julia Horak[1], Victoria Martin[1], Taru Sandén[2], Heide Spiegel[2]
AFFILIATIONS
[1]Centre for Microbiology and Environmental Systems Science, University of Vienna, Austria
[2]Department for Soil Health and Plant Nutrition, Austrian Agency for Health and Food Safety (AGES),
Vienna, Austria
CORRESPONDING AUTHOR: Jörg Schnecker
email: joerg.schnecker@univie.ac.at
ABSTRACT
Microbial respiration, growth and turnover are driving processes in the formation and decomposition of
soil organic matter. In contrast to respiration and growth, microbial turnover and death currently lack
distinct methods to be determined. Here we propose a new approach to determine microbial death rates
and to improve measurements of microbial growth. By combining sequential DNA extraction to
distinguish between intracellular and extracellular DNA and $^{18}O$ incorporation into DNA, we were able to
measure microbial death rates. We first evaluated methods to determine and extract intracellular and
extracellular DNA separately. We then tested the method by subjecting soil from a temperate agricultural
field and a deciduous beech forest to either 20 ℃, 30 ℃ or 45 ℃ for 24 h. Our results show, that while



mass specific respiration and gross growth either increased with temperature or remained stable,
microbial death rates strongly increased at 45 ℃ and caused a decrease in microbial biomass and thus in
microbial net growth. We further found that also extracellular DNA pools decreased at 45 ℃ compared to
lower temperatures, further indicating enhanced uptake and recycling of extracellular DNA along with
increased respiration, growth and death rates. Additional experiments including soils from more and
different ecosystems as well as testing the effects of factors other than temperature on microbial death
are certainly necessary to better understand the role of microbial death in soil C cycling. We are
nevertheless confident that this new approach to determine microbial death rates and dynamics of
intracellular and extracellular DNA separately will help to improve concepts and models of C dynamics in
soils in the future.

1 INTRODUCTION
Microorganisms are the driving force that sustains the 1450 Gt carbon (C) in soils globally (Liang et al.,
2017; Scharlemann et al., 2014). Active microorganisms take up and convert plant derived C and soil
organic C into microbial biomass and release C as $CO_2$ to the atmosphere via respiration. Upon cell death,
microbial C is released back to the soil solution and can be stabilized on mineral surfaces or in aggregates.
While causes for microbial death in soils can be numerous, ranging from osmotic shock and dehydration
to viral lysis and predation (Sokol et al., 2022), the relevance of this process and of the microbial
necromass pool for soil C cycling is undisputed. Since a large proportion of SOM is passing through the
microbial biomass pool (Kallenbach et al., 2016; Miltner et al., 2012), the process of microbial death might
be of equal importance as microbial growth for SOM formation.
Methodological developments in the last decades have made it possible to measure microbial C uptake
(Bååth, 2001; Frey et al., 2013; Rousk and Bååth, 2007). Substrate independent methods, that use [18]O
have enabled the measurement of growth of the whole soil microbial community and individual taxa

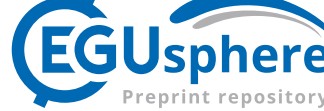

without changing substrate availability for microbes (Blazewicz and Schwartz, 2011; Hungate et al., 2015;
Spohn et al., 2016). Recently developed methods even allow these measurements without changing soil
water contents (Canarini et al., 2020; Metze et al., 2023). In contrast to uptake and growth, turnover and
death rates of the microbial community have not seen a suitable method yet. Microbial turnover can be
calculated using only growth rates and the microbial biomass pool (e.g., Prommer et al., 2020; Spohn et
al., 2016), under the assumption of a stabile state of the microbial community and no net changes in the
living microbial biomass as well as death rates being the same as growth rates. An assumption that might
not always be met under natural conditions.
A reason for the lack of methods to determine microbial death rates might be that DNA extractions used
for $^{18}$O-based methods do not account for extracellular DNA (eDNA). Extracellular DNA is DNA that persists
outside of intact microbial cells (Pietramellara et al., 2009). The eDNA pool is on the one hand fed by
disintegrated microbial cells (Ascher et al., 2009; Nagler et al., 2020), which could have died as
consequence to chemical or physical stressors or lysis caused by predators or viruses (Sokol et al., 2022).
On the other hand, it has been shown that DNA is actively exuded by microorganisms as an integral
component of microbial biofilms in soils (Cai et al., 2019; Das et al., 2013). eDNA can be rather prominent
in soils and has been shown to account for up to 80 % of the total DNA extracted (Carini et al., 2016). Such
a large pool of DNA, irrespective of its origin has the capacity to mask subtle changes in the pool of DNA
inside living microbial cells (iDNA) and to bias measurements of microbial growth that are based on the
determination of DNA contents.
Here we propose a novel approach to assess microbial turnover rates. We suggest that separating the
eDNA and iDNA pools upon the determination of microbial growth rates based on $^{18}$O-water incorporation
into DNA harbors several advantages over the conventional method. The adaptation provides more
precise growth rate measurements as it also allows the calculation of only iDNA production rates.
Accordingly, changes in the iDNA pool can be used to calculate gross DNA release rates, i.e. microbial



death rates. Besides providing insights into microbial death rates, observing changes in the iDNA as well
as eDNA pools holds potential information about microbial processes like microbial DNA uptake and
recycling.
In addition to evaluating extraction methods for eDNA and iDNA and evaluation of $^{18}O$ incorporation in
the two DNA pools over time, we have tested the method by subjecting soils to different temperatures.
We used 20 ℃, 30 ℃ and 45 ℃ assuming that these temperatures represent three distinct but relevant
temperatures for microbial activities in the investigated soils. The investigated soils were from two
contrasting temperate systems (an agricultural field and a deciduous forest) that regularly experience 20
℃ and sometimes even 30 ℃ in the topsoil layers (Schnecker et al., 2022). Around 30 ℃ is the assumed
optimum temperature for microbial activity for microorganisms in many soils (Birgander et al., 2018;
Nottingham et al., 2019; Rousk et al., 2012) and 45 ℃ is a temperature, that has been shown to be beyond
the temperature optimum where microbial process rates are reduced in comparison to under 30 ℃ (Cruz-
Paredes et al., 2021; Rousk et al., 2012). We expected, that (1) mass specific respiration, would increase
from 20 ℃ to 30 ℃ and further to 45 ℃. We further hypothesized that (2) a previously shown decrease
in microbial net growth above the temperature optimum at 30 ℃ would be caused by increased microbial
death and a net decrease in microbial biomass.

2 MATERIALS AND METHODS
2.1 Sampling sites
Soil samples were collected from an agricultural field site and a deciduous forest. The long-term
agricultural field experiment near Grabenegg, in Alpenvorland, Austria (48°12´N 15°15´E), was established
in 1986 and previously described in Spiegel et al. (2018). The soil is classified as gleyic Luvisol (Spiegel et
al., 2018) and has a silt loam texture (10 % sand, 73 % silt, and 17 % clay). Soil pH is 6.1 (Canarini et al.,
2020). The forest study site at the experimental forest Rosalia, Austria (47°42'N, 16°17'E) is dominated by



European beech (*Fagus sylvatica* L.). The soil at the site is a gleyic Cambisol (Leitner et al., 2016). Texture
is a sandy loam (55 % sand, 38 % silt, and 7 % clay), soil pH is 4.9 (Canarini et al., 2020). Soils were sampled
from 0-5cm depth with a soil corer with a diameter of 2 cm. At both sites, 10 soil cores per each of the
four replicate plots were combined to one sample resulting in four field replicates per site. At the
agricultural site, the four sampled plots were 7.5 m wide and 28 m long and at least 5 m apart from the
next plot. At the forest site, the 3 m by 3 m plots were at least 10 m apart from each other. All samples
were homogenized by sieving in the field through a 2 mm mesh before they were transported to the
laboratory.
2.2 Experimental setup
To evaluate the feasibility of eDNA extraction and determination of eDNA pool size, as well as the potential
for its use in conjunction with $^{18}$O-based determination of microbial growth, we carried out three tests.

109         1)  Comparing methods to collect or remove eDNA

110         2)  Dynamics of eDNA over time at constant temperature

111         3)  Temperature response of microbial biomass, DNA pools, microbial growth, death, and respiration

2.2.1   Comparing methods to collect or remove eDNA
To determine the contribution of eDNA to the total DNA pool, we compared two published methods. The
first method removes eDNA by addition of DNases (DNase method, (Lennon et al., 2018)), the second
method is based on a sequential DNA extraction (Ascher et al., 2009).
For this test, soil samples were collected in October 2021 and kept at 4 ℃ for one week before the
experiment. For the DNase method, 400 mg of field moist soil were weighed in two 2 mL plastic tubes
each. All tubes were then amended with 440 μL buffer consisting of 382.5 μL of ultrapure water, 5 μL of
1 M MgCl$_2$, 2.5 μL of bovine serum albumin (10 mg/ml), and 120 μL of 0.5 M Tris-HCl (pH 7.5). One of the
two samples further received 40 μL DNase I solution (10U/μL), the other tube received 40 μL ultrapure
water and served as control. Both samples were incubated in an incubator at 37 ℃ for 1 h. Afterwards 25



μL 0.5M EDTA was added, and the tubes were transferred to an incubator at 75 °C to stop DNase activity.
After 15 min, the samples were centrifuged, the supernatant was discarded, and the remaining sample
was extracted using FastDNA™ SPIN Kit for Soil (MP Biomedicals).
For the sequential DNA extraction, we used the chemicals and materials provided in the FastDNA™ SPIN
Kit for Soil (MP Biomedicals). For this approach 400 mg of field moist soil were weighed in the 2 mL Lysing
Matrix E tubes from which the contents had been emptied and collected in a 2 mL plastic vial. We added
1100 μL sodium phosphate buffer to the soil in the lysing tube and shook the vials gently in a horizontal
position at 100 rpm at 4 °C for 20 minutes. After this, the vials were centrifuged at 12500 rpm for 2 min
and the supernatant was collected as the eDNA containing fraction. The original content of the Lysing
matrix E tubes was returned to the tubes and handled as described in the manufacturer instructions to
obtain the iDNA pool. To the eDNA-fraction we then added 250 μL Protein precipitation solution and
followed the MP bio instructions after this step, except for additional centrifugation steps for separating
binding matrix and the liquid solution. After DNA extraction and purification, DNA extracts were stored at
-80C until further use. In addition to these two approaches, the same soils were also extracted regularly
using the FastDNA™ SPIN Kit for Soil (MP Biomedicals) to determine the total extractable DNA pool. The
DNA concentration of all extracts was determined fluorometrically by a Picogreen assay using a kit (Quant-
iT™ PicoGreen® dsDNA Reagent, Life Technologies). Content of eDNA determined with the DNase method
was calculated by subtracting the DNA content of samples that received DNase I from samples that only
received water and served as control.

2.2.2   Dynamics of eDNA over time at constant temperature
In this experiment, we explored the changes in eDNA and iDNA pools over time as well as the
incorporation of $^{18}O$ from added water into these two distinct DNA pools. Soils were sampled in August
2022 and the incubation was started one week later, where samples were stored at 20 °C. For the



experiment, 400 mg of field moist soil were weighed into empty lysing matrix E tubes and amended with
$^{18}$O-water to achieve 60 % of the soils water holding capacity and a labelling of 20 atom percent (atm %)
of the total water in the soil. From each of the four field replicates, 7 vials were filled, labelled with $^{18}$O
water and closed. Immediately after label addition and after 6 h, 12 h, 24 h, 48 h, 72 h and 168 h, eDNA
and iDNA was extracted with sequential DNA extraction as described above. DNA concentrations in all
DNA fractions were determined using the Picogreen assay. Subsequently, total oxygen content and $^{18}$O
enrichment of the purified DNA fractions were measured following Spohn et al. (Spohn et al., 2016) and
Zheng et al. (Zheng et al., 2019) using a thermochemical elemental analyzer (TC/EA, Thermo Fisher)
coupled via a Conflo III open split system to an isotope ratio mass spectrometer (Delta V Advantage,
Thermo Fisher).

2.2.3    Temperature response of microbial biomass, DNA pools, microbial growth, death and respiration
In this experiment we subjected the samples to three different temperatures to test the response of
microbial communities. Soils were collected in August 2022 and stored at 20 ℃ for two days before the
start of the experiment.
For the incubation, around 400 mg of soil were weighed into empty lysing matrix E tubes. From each field
replicate, five lysing matrix E tubes were filled. Two sets of samples were amended with natural
abundance water and three sets were amended with $^{18}$O-water to achieve 60 % water holding capacity
and 20 atm % $^{18}$O in the final soil water, when $^{18}$O-water was added. One set of samples that received
natural abundance water was extracted immediately using sequential DNA extraction. The second set of
natural abundance samples and one set of samples with $^{18}$O-water were put in an incubator set to 20 ℃.
A second set was put in an incubator set to 30 ℃ and the third set of samples was incubated at 45 ℃.
After 24 h in the incubators, all samples were subjected to sequential DNA extraction to recover eDNA
and iDNA pools. All obtained DNA extracts were stored at -80 ℃ before DNA concentrations were



determined using Picogreen assay and oxygen content and $^{18}O$ enrichment were determined as described
above.
In addition to the $^{18}O$-incubation, we determined microbial respiration rates and microbial biomass C
following the descriptions in Schnecker et al. (Schnecker et al., 2023). For microbial respiration 400 mg of
soil were weighed in plastic vials, water was added to achieve 60 % WHC and the open plastic vials
containing the soil were inserted into 27 mL headspace vials. The headspace vials were sealed with a
rubber septum. This was done in three replicates for each soil sample, with one set being incubated at 20
℃, 30 ℃ and 45 ℃ respectively. In addition to the headspace vials containing soil samples, 5 empty glass
vials were sealed with rubber septa and added for each temperature. After 24 h, we measured the $CO_2$
concentration in the headspace vials by taking gas samples from a sealed headspace vial and measured it
directly with an infrared gas analyzer (EGM4, PP systems). Microbial respiration rate was then calculated
as the difference in $CO_2$ concentrations between the vials containing soil samples and empty glass vials,
which contained the air at the start of the incubation. The net increase in $CO_2$ was divided by the
incubation time.
Microbial biomass C (MBC) was determined following an approach based on (Brookes et al., 1985) and
described in Schnecker et al. (Schnecker et al., 2023) with parallel determinations for MBC at the three
temperatures. MBC was determined in 1M KCl and measured on a TOC/TN analyzer (TOC-L CPH/CPN,
Shimadzu). Measured MBC values were divided by 0.45 (Wu et al. 1990) to account for extraction
efficiency.

For each of the three temperatures, we calculated microbial gross growth rates (gG), microbial net growth
rates (nG), microbial gross death rates ($DNA_{death}$) and microbial carbon use efficiency (CUE).
Microbial gross growth was calculated following Canarini et al (Canarini et al., 2020) as the amount of
iDNA produced:



$$iDNA_{produced} = O_{iDNA\,extr} * \frac{^{18}O\,at\%_{iDNA\,L} - ^{18}O\,at\%_{iDNA\,n.a.}}{^{18}O\,at\%_{soil\,water}} * \frac{100}{31.21}$$
Where $O_{iDNA\,extr}$ is the total amount of oxygen in the iDNA extract, $^{18}O\,at\%_{iDNA\,L}$ and $^{18}O\,at\%_{iDNA\,n.a.}$ are the
$^{18}O$ enrichment in the labeled DNA extracts from the different temperatures and unlabeled DNA extracts
respectively, and $^{18}O\,at\%_{soil\,water}$ is the $^{18}O$ enrichment of the soil water. The fraction at the end of the
formula accounts for the average oxygen content of DNA (31.21%, (Canarini et al., 2020; Zheng et al.,

199 2019)).

Mass specific gross growth rate (MSgG) was calculated by dividing $iDNA_{produced}$ by the amount of iDNA in
the respective sample.
Microbial net growth rate was calculated by subtracting the amount of iDNA in the samples that were
extracted immediately from the amount of iDNA at the end of the incubation divided by the incubation
time. Mass specific net growth rate (MSnG) was calculated by dividing nG by the iDNA content at the end
of the incubation. Microbial gross death rates were calculated by using the following formula:
$$DNA_{death} = |\,\Delta iDNA\; - iDNA_{produced}\,|$$

Where microbial death rates ($DNA_{death}$) are determined by subtracting iDNA growth ($iDNA_{produced}$),
determined by $^{18}O$ incorporation into iDNA, from the net growth rate ($\Delta iDNA$). Mass specific gross death
(MSD) was calculated by dividing $DNA_{death}$ by the iDNA content.
Microbial CUE was calculated using the following equation (Manzoni et al., 2012):
$$CUE = \frac{C_{Growth}}{C_{Growth} + C_{Respiration}}$$
Where microbial biomass C produced ($C_{Growth}$) during the incubation was calculated as $iDNA_{produced}$ divided
by the total amount of iDNA in the sample and multiplied by MBC values. Microbial respiration ($C_{Respiration}$)



was calculated from the respiration measurements described above. Mass specific microbial respiration
(MSR) was calculated as $C_{Respiration}$ divided by MBC.

2.3 Statistics
All statistical analyses were performed in R 4.1.2 (R Development Core Team, 2013). To determine
whether eDNA or iDNA pools or $^{18}O$ atom percent access were different from timepoint 0 in Experiment
2.2.2 we used two sample comparison tests. We used either t-tests, Welch t-tests when variances were
not homogeneous or Wilcoxon rank sum tests when data were not normally distributed. We used Fit
Linear Model Using Generalized Least Squares (R function 'gls') and Linear Mixed-Effects Models ('lme'),
which are both contained in the R package 'nlme' (Pinheiro et al., 2021) and Estimated marginal means
('emmeans') to determine effects of temperature on microbial processes and MBC and DNA pools
(Experiment 4) and differences in the extraction assays (Experiment 2.2.1). To account for non-normal
distributed residuals, we used log transformations where necessary. If residuals of the models were non-
homoscedastic, we introduced weights in the respective functions. We also introduced field plots as
random effects. Different models including weights and random effects were set up and compared with
the ANOVA('anova'). If models were statistically different, we chose the model with the lowest Akaike
information criterion (AIC). Statistical tests were assumed to be significant at $p<0.05$.

3 RESULTS and DISCUSSION
3.1 Comparing methods to collect or remove eDNA
To distinguish eDNA and iDNA, we tested two methods. First, eDNA digestion by DNase (Lennon et al.,
2018) and sequential extraction (Ascher et al., 2009). Compared to regular DNA extraction, sequential
extraction yielded on average 23.1 % less and the DNase method yielded on average 78.2 % less total DNA
(Table 1). The DNase digestion also did not work as expected in two out of four replicates at each site.



Table 1 Comparison of methods to estimate eDNA in soil samples from two soil systems.

|  | agricultural soil | | | | forest soil | | | |
|---|---|---|---|---|---|---|---|---|
|  | mean | min. | max. | n | mean | min. | max. | n |
| regular DNA extraction, total DNA (µg DNA g⁻¹ dry soil) | 6.791 | 6.060 | 7.285 | 4 | 19.67 | 13.32 | 22.50 | 4 |
| sequential DNA extraction, total DNA (µg DNA g⁻¹ dry soil) | 4.956 | 4.556 | 5.190 | 4 | 15.91 | 12.53 | 19.69 | 4 |
| DNase method, total DNA (µg DNA g⁻¹ dry soil) | 0.756 | 0.712 | 0.805 | 4 | 6.388 | 5.460 | 6.830 | 4 |
| Sequential DNA extraction, eDNA (% of total) | 2.447 | 1.838 | 3.265 | 4 | 6.472 | 5.957 | 7.183 | 4 |
| DNase method, eDNA (% of total DNA) | -7.063 | -32.19 | 15.14 | 4 | -6.917 | -30.14 | 7.024 | 4 |
| DNase method, eDNA (% of total DNA), corrected for negative values | 10.60 | 6.061 | 15.14 | 2 | 6.053 | 5.082 | 7.024 | 2 |


Due to these findings and the fact, that the DNase method uses incubation temperatures of 35 ℃ and 75
℃, which likely interfere with potential temperature treatments, we decided to use sequential extraction
for our further experiments. Sequential extraction also has the advantage that both eDNA and iDNA are
recovered and can be used for further analyses. The amounts of eDNA recovered with sequential DNA
extraction were on average 2.4 % of total DNA in agricultural soils and 6.5 % of total DNA in forest soils,
which is on the lower end of the range found in other studies (Carini et al., 2016; Lennon et al., 2018).

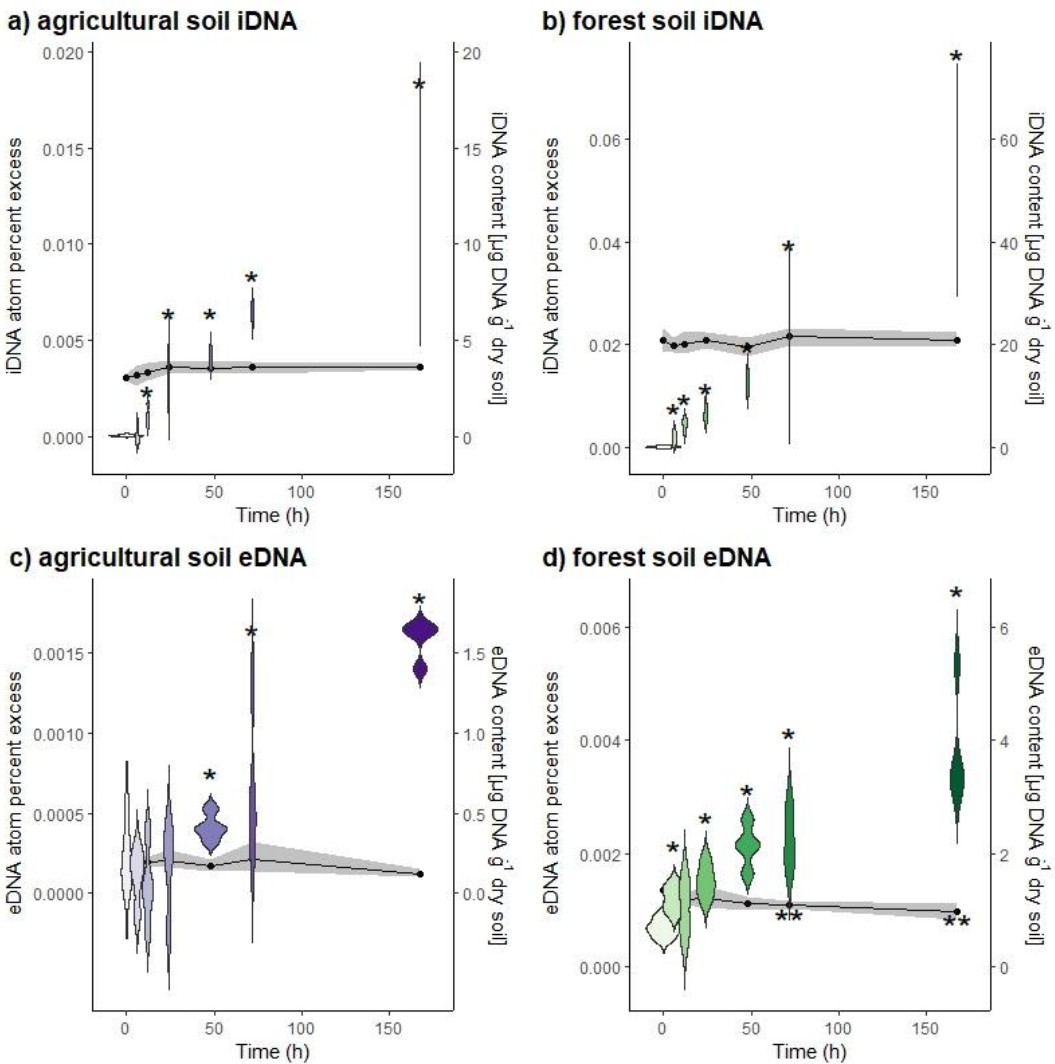

Figure 1. Temporal development of DNA pools and $^{18}$O enrichment during incubation with $^{18}$O-water.

Upper panels depict iDNA pools and enrichment in a) agricultural soils and b) forest soils. Lower panels

depict eDNA pools and enrichment in c) agricultural soils and d) forest soils. Violin plots represent $^{18}$O

enrichment of DNA pools (atom percent excess) and dot and line plots DNA pool sizes over time. Asterisks

indicate significant differences (p-value < 0.05) from timepoint 0.



We also determined the change in eDNA and iDNA content as well as the incorporation of $^{18}O$ from
amended $^{18}O$-labelled water into these two DNA pools over time (Figure 1). We found that only the
amount of eDNA in forest soils slightly decreased over time and was significantly lower after 72 h and
after 168 h compared to the initial eDNA content (Figure 1d). In forest soils, the iDNA content and both
DNA pools in the agricultural soil did not change over time (Figure 1 a-c). The amended $^{18}O$ was
incorporated into both DNA pools at both sites over time, indicating production of iDNA and eDNA. While
we could detect $^{18}O$ label at the latest after 12 h in both DNA pools of the forest soil and the iDNA pool of
the agricultural soil, increased $^{18}O$ values could only be found after 48 h in the eDNA pool of the
agricultural soil. This could indicate, that the eDNA pool in the agricultural soil might mainly be fed by
microbial death, and that the $^{18}O$ is thus first incorporated in iDNA and only when these newly formed
cells die, the label is released as eDNA. In the forest soil our findings indicate that eDNA is actively exuded
from the beginning on. If eDNA is actively exuded as e.g. part of microbial biofilm (Das et al., 2013; Nagler
et al., 2018; Pietramellara et al., 2009) depends on the present microorganisms (Cai et al., 2019). The
amount of eDNA produced can also vary for different microorganisms (Figure S1).

3.2. Temperature response of microbial biomass, DNA pools, microbial growth, death, and respiration
To test the combination of sequential DNA extraction and $^{18}O$ incorporation in DNA, we subjected soil
from the agricultural site and the forest site to three different temperatures. Microbial processes and
activity have been shown to strongly increase with temperature up to a temperature optimum (Rousk et
al., 2012). Above this temperature threshold conditions are adverse and have been shown to lead to a
reduction of the microbial biomass (Riah-Anglet et al., 2015). By subjecting the two investigated soil types
to 20 ℃, 30 ℃ and 45 ℃ we found that MBC was not affected by temperature (Figure 2 a,b). The content
of iDNA did not change from 20 ℃ to 30 ℃ and decreased significantly when soils were brought to 45 ℃
(Figure 2 c,d). The decrease in iDNA at 45 ℃ indicated that a part of the microbial community died because




of the high temperature and DNA might have been lost from within the microbial cells. In agricultural soils,
eDNA contents were significantly lower at 30 ℃ and 45 ℃ than at 20 ℃, while eDNA contents in forest
soils only dropped significantly in the 45 ℃ treatment (Figure 2 e-f). We suggest that decreasing eDNA
contents with temperature rather indicate a higher degradation and recycling of eDNA than the reduction
of eDNA release from microbial cells.

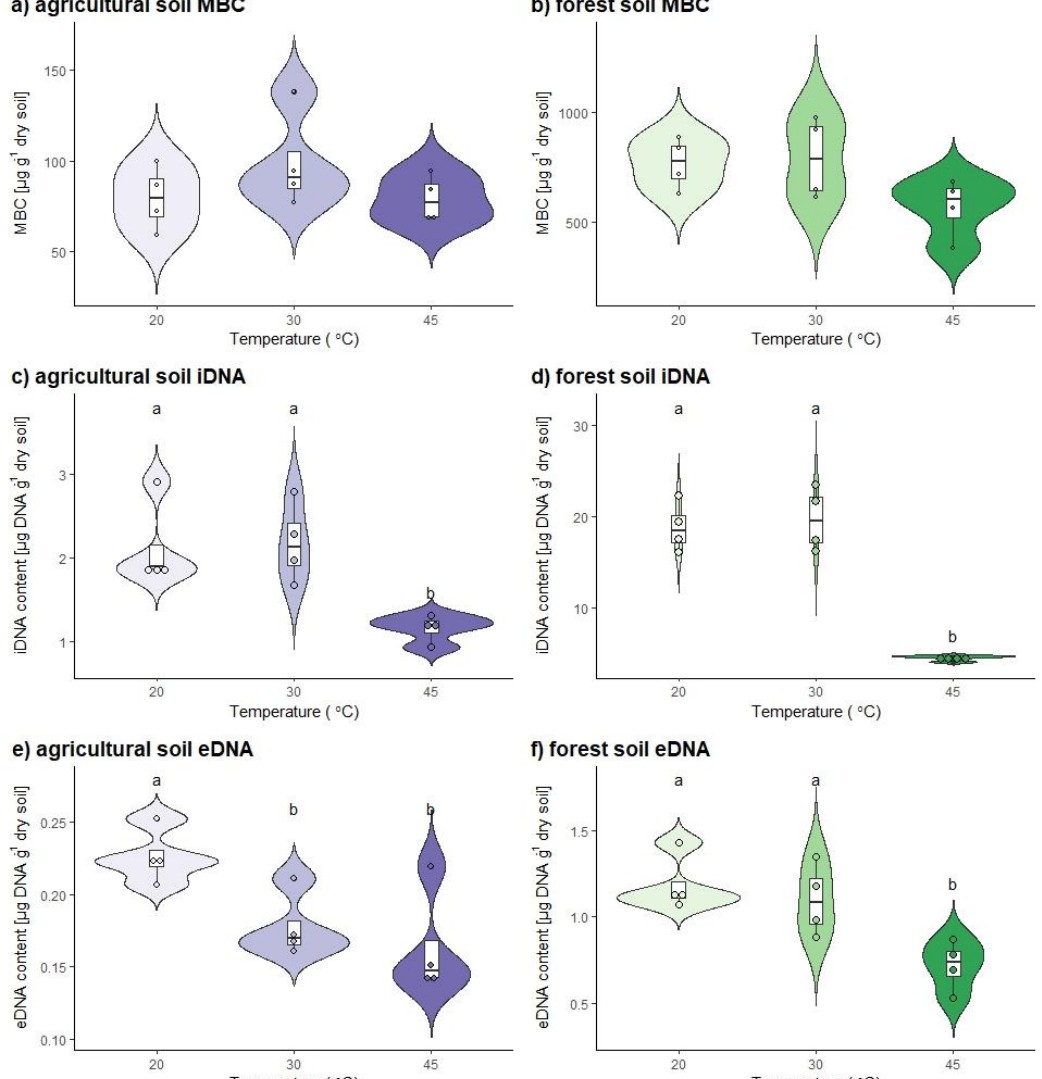






Figure 2. Microbial pool sizes in the two investigated soils after incubation at three different temperatures for 24 h. Results for agricultural soils are shown on plots a, c and e. Forest soils are shown in plots b, d, f. Microbial biomass C is shown in a) and b), iDNA contents are shown in c) and d) and eDNA contents are shown in e) and f). Statistically significant differences between pool sizes at the three investigated temperatures are marked with different letters above the violin plots.

Mass specific respiration increased in both soils from 20 ℃ over 30 ℃ to 45 ℃ (Figure 3 a) confirming previous findings of other studies (Birgander et al., 2018; Cruz-Paredes et al., 2021; Rousk et al., 2012). Mass specific gross growth did not change with temperature in agricultural soils but increased from 20 ℃ to 30 ℃ and even to 45 ℃ in forest soils (Figure 3 b). This is in contrast to previous studies (Birgander et al., 2018; Cruz-Paredes et al., 2021; Rousk et al., 2012), which found that microbial uptake of leucine in microbial biomass and acetate in fungal ergosterol, which was used as indicators of growth, showed a clear temperature optimum around 30 ℃ and concomitant decrease at higher temperatures. These studies however used other methods than we did under the assumption of no net decrease in microbial biomass and equal rates of microbial growth or uptake and microbial death. While our data also show no mass specific net change in microbial biomass from 20 ℃ to 30 ℃, a significant negative mass specific net growth was observed at 45 ℃ in both soils (Figure 3 c). When we combine MSgG and MSnG the calculated microbial death rates were significantly higher at 45 ℃ than at 20 ℃ and 30 ℃ in both soils (Figure 3 d). Carbon use efficiency decreased with increasing temperature in forest soil, while it stayed constant in agricultural soils (Figure 3 e). This finding adds to an ever-growing list of ambiguous reactions of CUE to soil temperature (e.g. (Hagerty et al., 2014; Schnecker et al., 2023; Simon et al., 2020; Walker et al., 2018)) and once again shows, that CUE should be used with caution to infer soil C cycling. As showcased in our experiment, CUE was low at high temperatures in forest soils while growth as well as death rates were high, thereby indicating fast microbial C cycling.

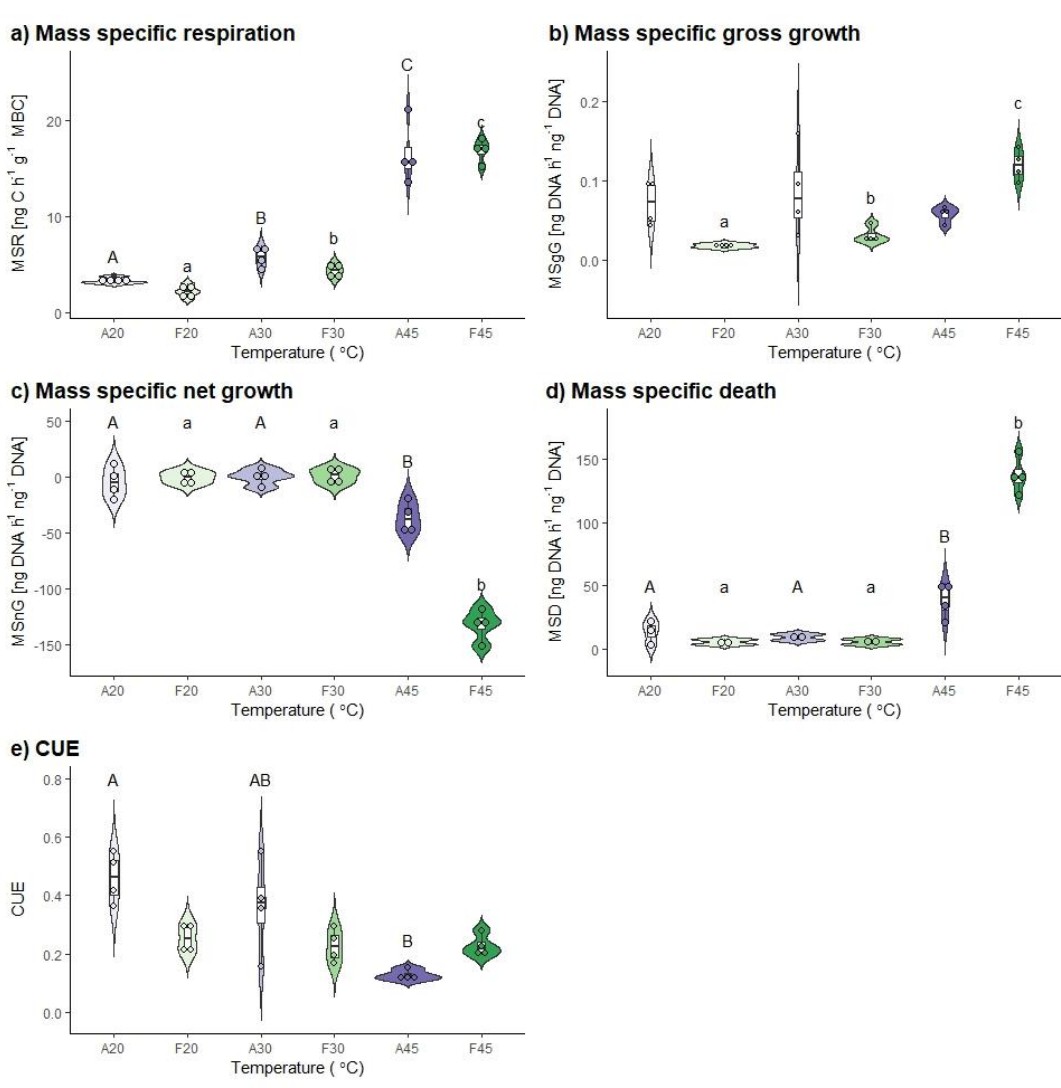

309

Figure 3. Mass specific microbial process rates and CUE in the two investigated soils after incubation at three different temperatures for 24 h. Results for agricultural soils are shown in purple hues and for forest soils are shown green hues. Statistically significant differences between pool sizes at the three investigated temperatures and respective soil are marked with different letters above the violin plots. Capital letters for differences between agricultural soils and lower-case letters are used to indicate differences for forest soil.



316

CONCLUSION

In conclusion we here present an approach to determine microbial death rates and turnover by accounting for eDNA dynamics. To our knowledge, this is the first time microbial death rates were investigated in addition to microbial growth rates and net changes in microbial iDNA. With this approach we could show that microbial respiration and microbial growth in the two investigated soils increase with temperature even up to 45 ℃, a temperature, that is considered to be way beyond the temperature optimum of most temperate microbial communities. The often observed drop in microbial growth or uptake at high temperatures was however caused by the death of a significant part of the microbial community and higher microbial death rates. While there is certainly room for improving the method and the necessity to investigate its feasibility in other soil systems and under different environmental conditions, we think that this approach will help to shed light on the role of microbial death in soil and a step forward to understand soil C cycling.

AUTHOR CONTRIBUTION

**Jörg Schnecker:** Conceptualization (lead); investigation (supporting); methodology (supporting); supervision (lead); formal analysis (lead); writing – original draft (lead) writing – review and editing (equal).

**Theresa Böckle:** investigation (equal); methodology (equal); writing – review and editing (equal). **Julia Horak:** investigation (equal); methodology (equal); writing – review and editing (equal). **Victoria Martin:** investigation (supporting); methodology (supporting); writing – review and editing (equal). **Taru Sandén:** resources (equal); writing – review and editing (equal). **Heide Spiegel:** resources (equal); writing – review and editing (equal).

COMPETING INTERESTS



The authors declare that they have no conflict of interest.
ACKNOWLEDGEMENT
This research was funded by the Austrian Science Fund (FWF TAI 328). We thank Sophie Zechmeister-
Boltenstern, University of Natural Resources and Life Sciences for granting access to the field site.

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

color).