# Peer review of "Improving measurements of microbial growth, death, and turnover by accounting for extracellular DNA"

_EGUsphere, 2023_

## Author Response (AR1)

Me and my co-authors would like to thank the topic editor as well as the two reviewers for their valued comments and suggestions. Below is a point-by-point response to these suggestions based on our answers during the interactive discussion. Most notably we have now included a caveats and opportunities section in the results and discussion section of the manuscript.

Kind Regards,

Jörg Schnecker

Topic Editor:

The manuscript by Schnecker and coauthors has now been carefully reviewed by two experts. Both reviewers agreed the that the manuscript's topic is of interest to the scientific community, but they differed in their assessment of the quality of the work. While Reviewer 1 considered the paper strong, Reviewer 2 had several concerns regarding the methodology, and consequently the validity of the conclusions drawn from the results. Given that soil microbiology is a field of imperfect methods, I have decided to grant the authors the opportunity to revise the manuscript and more explicitly clarify the methodological limitations. Additionally, I recommend the authors take care in making improvement to the manuscript to address the reviewers concerns a thoroughly as possible.

From my own read of the paper, I found the approach to measuring eDNA production to be interesting and potentially useful as an indicator of microbial necromass and extracellular polymeric substance production. In my opinion, this advance is arguably more novel than the possible improvement to estimates of microbial death achieved by accounting for eDNA and could be highlighted in the revision.

>>We have added a section dealing with caveats and opportunities (L330-L344)

R2:

Overall, this manuscript seeks to advance understanding of microbial growth, death, and turnover - all significant topics highly relevant to many research areas (e.g. soil ecology, biogeochemistry, climate modeling). However, I have some concerns about the methods presented and think more evidence is needed to validate the sequential DNA extraction protocol.

I agree that a method to separate extracellular DNA (eDNA) and intracellular DNA (iDNA) during DNA extraction of soil would be a great advancement in the field, though I am not convinced that the sequential DNA extraction protocol presented here has achieved this. To be fair, I don't believe the original protocol adequately demonstrated the separate extraction of eDNA and

iDNA and while it is not the authors' obligation to address these weaknesses a better validation of this method would be of great benefit to this manuscript and to the field in general.

>> We here aimed to combine sequential extraction of eDNA and isotopic measurements of microbial growth. We are well aware that both methods individually have their inherent issues and biases. By combining the two methods we had to accept further tradeoffs between how practical our method would be and how precise it is. Despite the caveats of the method, and we have mentioned that there is room to improve in the concluding section of the manuscript, we think that our approach is a good starting point for investigations into microbial death and turnover in the future. To improve the manuscript, we have now included a whole section in the Results and Discussion section on the caveats and opportunities of the presented approach (L330-344).

One concern is that eDNA in the original sequential DNA extraction protocol (Ascher et al., 2009) was recovered by gently shaking vials of soil in sodium phosphate buffer repeated 3 times; however, this step was only performed once in this study. While I don't believe that 3 is a magic number, do the authors have any evidence that all eDNA was removed in 1 'wash' step? It would be more convincing if this was presented (or a quantitative estimate). Without confirmation that all eDNA was removed in the initial step, I am wary of the successive measurements of the iDNA, which almost all subsequent calculations of growth and death are based upon.

>> As mentioned above, we had to make tradeoffs between practical and precise aspects. While we did not repeat the shaking procedure, we instead increased the time of extraction compared to the original manuscript. Based on the findings of our experiment we also think that we could extract the active part of the eDNA pool. We saw that eDNA is only labelled after a certain time in agricultural soils indicating that 18O was first incorporated into new DNA inside the cell and only later released and extracted as eDNA. We also saw that our eDNA pools were susceptible to soil temperature. We also found comparable proportions of eDNA with sequential extraction and DNase methods when the DNase method worked.

Furthermore, it looks like there are significant differences in the amount of DNA recovered from the soils using the 3 methods described (regular extraction, sequential extraction, and DNAse method), though no statistics are shown, and this discrepancy is not discussed.

>> We have added a discussion on this topic in the manuscript L238-242. We think that the lower yields using sequential extraction arise from the additional pipetting steps and the separate cleaning steps for eDNA and iDNA. We know that the used extraction kit does not have 100% extraction efficiency and some DNA is lost during the extraction process. This has been discussed in Pold et al 2020. Investigations in extraction efficiencies would for sure improve the absolute numbers determined whit the presented approach but is beyond the scope of this manuscript.

While I think this method has potential and find the approach exciting, I think more evidence supporting the idea that eDNA and iDNA are truly being separately quantified is needed for interpretations about microbial growth and death inferred from iDNA. If the authors can address this, I believe this manuscript presents a significant advancement in the field.

>> We agree that there is room to improve the method. However, our approach is still valid in comparing the response of individual soils to different treatments e.g. different soil temperatures as we did in this manuscript. We further aimed do provide our approach early on to the scientific community so it can be tested e.g. in different soils and improved (see L329-343).

General comments

The materials and methods section for experiment 2.2.3 has numerous of acronyms; however, most of them are not used more than twice (exceptions include MBC and CUE). I suggest removing the abbreviations for terms not used more than twice (e.g. microbial gross growth rates (gG), microbial net growth rates (nG), mass specific gross growth rate (MSgG), etc.) to improve clarity.

>> We have removed nG and gG but decided to keep the other abbreviations because we also used them in the graphs. We further removed the abbreviations from the text outside of the methods section to increase readability.

Specific comments

L146-148: Can the authors discuss what the moistures of the 'field moist' soils were and how much water was added in order to achieve 60% water holding capacity? In other words, were the agricultural and forest soils similarly wet or was there a significant difference in the amount of 18O water added to the two soil types? If one of the soils was relatively dry, a birch effect could potentially create a nutrient pulse in the different ecosystems, influencing growth and data interpretation.

>> Field soils were moist to the touch. Field fresh soils were at 28% WHC for agricultural soil and 20% for forest soils. Therefore we do not think that birch effects have strongly influenced our results. Also eDNA content would probably have gone down if the microorganisms in the soil would have been strongly affected by re-wetting over the 7 days of incubation in exp 2.

L162-164: same comment as above.

L236-244: An average of 23% less DNA was recovered from the sequential extraction and 78.2% less total DNA was recovered from the DNAse method. In contrast, Ascher et al. (2009) found that the eDNA + iDNA quantities were always greater than the regular extraction protocol. Can you discuss/hypothesize in more detail why there were differences in recovered DNA from the 3 different methods?

>>Please see our comments above. We have included more discussion on this topic L238-242.

L239 - Table 1: Can the authors provide statistics comparing the amount of DNA recovered from the different methods?

>> We included statistics for the comparisons of total DNA recovered with the three different methods in Table 1.

L239 - Table 1: Can the authors discuss why negative values were reported for the DNase method? How was the DNase method, eDNA (% of total DNA) corrected for negative values?

>>Yes. The DNase method uses two aliquots of the same sample. One with DNAse and the other without DNase but instead the same amount of water. Both samples were incubated in an incubator at 37 ℃ for 1 h. Afterwards 25 µL 0.5M EDTA was added, and the tubes were transferred to an incubator at 75 ℃ to stop DNase activity. After 15 min, the samples were centrifuged, the supernatant was discarded, and the remaining sample was extracted using FastDNA™ SPIN Kit for Soil (MP Biomedicals). Content of eDNA determined with the DNase method was calculated by subtracting the DNA content of samples that received DNase I from samples that only received water and served as control. (see the methods section in the manuscript)

Unfortunately the DNA digestion did not work in 2 out of 4 field replicates for both of our sites which resulted in slightly higher values for the samples that received water than those that received DNase which resulted in negative values.

We corrected for these negative values by not considering them as shown in table 1 and indicated by different n. We will change the wording in the table from "DNase method, eDNA (% of total DNA), corrected for negative values" to "DNase method, eDNA (% of total DNA), excluding negative values". (see Table 1)

L245-246: The eDNA values do seem quite low. Please discuss possible reasons the eDNA was so much lower than other studies?

>> eDNA values were in our study indeed at the lower end of the range of previously reported values. Carini et al 2016 investigated 31 different soils across the USA including ecosystems from prairie to alpine tundra the eDNA contents ranged from 0 to 80% of total DNA for bacteria and fungi. Amongst the systems with the lowest eDNA percentages were agricultural soils. We have added a statement in the manuscript. (L252-255)

It also has to be noted that soils used in our experiments were not fresh but were kept for some days to a week before we used the for our experiments. In this time eDNA could have been decomposed or recycled.

L264: I don't know if finding 18O label in eDNA after 12 hours conclusively indicates that eDNA is exuded. There could be several possible reasons including an artifact of the method (e.g. there are more sensitive cells in forest soils that are lysed during the eDNA wash?) or maybe faster growth and death in the forest soils.

>>We do think that our findings support this statement. In agricultural soils we found that label recovery in eDNA lagged behind label recovery in iDNA. In forest soils this lag was much shorter. Faster growth or death rates are unlikely since mass specific growth and death rates as determined in the third experiment did not differ strongly between agricultural and forest soils. We have also seen in our culture experiment presented in the supplement, that some bacterial strains exude eDNA from the beginning on in large quantities. More sensitive cells might be a possibility we would however expect that the microbial cells in the fungal dominated forest soil would be sturdier than the rather bacteria dominated agricultural soil. We have added this as a possibility in the manuscript. (L275-276)

L275: I find that MBC not being affected by temperature surprising. Especially with the significant decreases in iDNA and eDNA measured. Can the authors discuss some possible explanations why MBC didn't change?

>> This was also surprising to us. The used incubation times were with 24h relatively short. We think that the MBC method is just not precise enough to capture the changes in MBC in this short time period.

R1:

This a great paper, I recommend to accept with minor revisions

General comments:

In this study, the authors aimed to assess and compare methods for extracting extracellular DNA (eDNA) in soil samples to better quantify microbial growth, death, and turnover. Two methods, eDNA digestion by DNase and sequential extraction, were tested. The DNase method resulted in a substantial reduction in total DNA (78.2%), and its effectiveness varied across replicates. Sequential extraction, on the other hand, yielded 23.1% less total DNA on average compared to regular DNA extraction. Following these findings, the researchers decided to use sequential extraction for further experiments due to its consistent performance and the potential interference of the DNase method with temperature treatments. Notably, the amounts of eDNA recovered with sequential extraction were found to be 2.4% in agricultural soils and 6.5% in forest soils, contributing novel insights to the field, especially considering the lower end of the range observed in other studies. The study also delved into the temporal

dynamics of DNA pools and 18O enrichment during incubation with labeled water. The results suggested that eDNA in agricultural soil might be primarily derived from microbial death, contrasting with the forest soil where eDNA appeared to be actively exuded from the beginning. This observation adds a significant dimension to our understanding of eDNA dynamics in different soil types. In the second part of the study, the researchers explored the temperature response of microbial processes, including biomass, DNA pools, and microbial activity. Notably, they found that the use of 18O incorporation in DNA combined with sequential DNA extraction allowed for a comprehensive analysis. The study revealed temperature-dependent changes in microbial biomass, iDNA, and eDNA contents, providing valuable insights into the ecological implications of temperature variations on microbial communities. Overall, this work presents a thorough investigation into eDNA extraction methods and the temperature effects on microbial processes, introducing new perspectives and highlighting the potential impact of these findings on our understanding of soil ecology. The combination of innovative techniques and insightful results contributes to the novelty and significance of this research. This work is a good starting point for future research to explore more soils with greater replication.

Specific comments:

**Introduction**

The introduction is clearly written. Below I suggest minor changes.

L53-56 – Run-on sentence consider rephrasing to improve readability.

>> We have split the sentence. L53-L57

L64 – Starting a sentence with eDNA is odd.

>> Changed to "Pools of eDNA…" L64

L76 – Some examples of this application would be nice (either here or later in the discussion)

>> We have added some thoughts on this in an extra section of the discussion. L330-L344

L82-85 – Sentence could be more concise to improve readability e.g. "Around 30 ℃ is considered the optimal temperature for microbial activity in many soils (Birgander et al., 2018; Nottingham et al., 2019; Rousk et al., 2012) , while 45 ℃ has been demonstrated to reduce microbial process rates compared to temperatures below 30 ℃ (Cruz-Paredes et al., 2021; Rousk et al., 2012).

>> We have changed the section to: "Around 30 ℃ is the assumed optimum temperature for microbial activity in many soils (Birgander et al., 2018; Nottingham et al., 2019; Rousk et al., 2012). At 45 ℃ microbial process rates are reduced in comparison to the temperature optimum at 30 ℃ (Cruz-Paredes et al., 2021; Rousk et al., 2012) L83-L85

**Methods**

Methods are thorough and descriptive, no significant changes requested.

L142 – Shouldn't this be 'Dynamics of eDNA *and iDNA* over time at constant temperature'?

 >> changed. L141

**Results and Discussion**

I provide the following suggestions in the hopes of improving the understandability of the results:

- The study notes a 23.1% loss of total DNA in sequential extraction compared to regular DNA extraction. It would be beneficial to explain whether this loss is likely proportional between eDNA and iDNA or if one pool is more affected than the other. Addressing the acceptability of a 23% DNA loss is crucial for understanding the reliability of the method.

  >> The loss of DNA is likely caused by the additional pipetting steps as well as the separate cleaning steps of iDNA and eDNA. While the soil extraction kits allow for relatively easy extraction and handling of large numbers of samples, we know that DNA is lost during the process. Ideally this extraction efficiency should be determined and accounted for in the calculations later on. We did not do this here. As stated in the conclusions, there are definitely aspects of this approach that can be improved. This is especially true for the extraction process. Our goal here was to adapt a published method to extract eDNA and combine it with commercially available DNA extraction kits for soils. We had to handle the tradeoff between easy applicability and precision. We further think that even if we very likely do not report the correct absolute numbers of eDNA or even iDNA content, we think that our approach is still valide to compare treatments like we did in the temperature experiment. We further aimed do provide our approach early on to the scientific community so it can be tested e.g. in different soils and improved (see L330-344).

- Acknowledge and discuss the observed greater extraction efficiency in forest soil compared to agricultural soil. Consider the potential implications for generalizing these results across different soil types and ecosystems. Discuss whether this discrepancy could lead to underestimating eDNA in certain soils and how this might influence the interpretation of results.

>> Not necessarily higher extraction efficiency but this might be dependent on soil properties or the microbial community composition. i.e. fungal:bacterial ratio or biofilm production vs not see supplementary experiment. We have added a statement in the manuscript (L330-344).

- Provide an interpretation for why eDNA in the studied soils comprises only 2-6% of the total, especially in comparison to the 80% figure mentioned in the introduction (L65). Discuss potential reasons for this discrepancy and the implications for the broader understanding of eDNA dynamics in different environments.

  >> 80% was the upper end measured in a wide range of soils. The range in Carini et al 2016 was from 0 to 80%. The soils used for our experiments were stored for at least a week after sampling. During this time the eDNA pool might have decreased. L252-255

- Consider adding a subheading for Section L254, such as "Dynamics of eDNA and iDNA Over Time at Constant Temperature," to improve the organization and clarity of the manuscript.
  >> We added the suggested subheading. L263

- L280-282 - Further elaborate on the conclusion that decreasing eDNA with temperature may indicate higher degradation rather than a reduction in the release of eDNA. Clarify the basis for this interpretation and discuss its significance in the context of microbial processes at varying temperatures.

  >>Since we do see an increase in Mass specific death rate and a concomitant decrease in the eDNA pool we concluded that the efflux from the eDNA pool must be increased. This could be eDNA uptake or degradation of eDNA. L292-296

- Discuss whether variations in temperature responses can be expected over time. Consider addressing potential temporal dynamics in microbial processes and eDNA release under different temperature conditions

  >>This is a very interesting question. Based on our data, we think however that interpretations in this direction would be too speculative and beyond the scope of this manuscript

- The study observes temperature-dependent changes in microbial biomass, iDNA, and eDNA, but the paper lacks a comprehensive discussion placing these findings in the broader context of existing literature. Providing a more thorough review of relevant studies and discussing how these results contribute to the field would enhance the manuscript

>> We checked the literature again and found that it is difficult to put our findings in relation to other studies investigation e.g. MBC pools in response to warming since most studies usually do not investigate such drastic temperature increases and high temperatures as we did here. We particularly used 45C to be sure to include a temperature that has negative effects on the present microorganisms. On the other hand, the studies which include such high temperatures often only report the measured growth and respiration rates, which we have discussed in the manuscript, but do not report e.g. MBC pools.

- Generalization of Findings: The study reports eDNA recovery percentages that fall within the lower end of the range found in other studies. However, the generalizability of these findings to different soil types, ecosystems, or geographical locations should be addressed. Discussing the potential variability and the factors influencing it would strengthen the study's applicability.

  >> Carini et al 2016 investigated 31 different soils across the USA including ecosystems from prairie to alpine tundra. In their study the authors could not find discernable connections between ecosystems and the amount of eDNA found. Based on the two investigated soils here we think it would go too far to generalize our findings for other soil types or ecosystems.

- Implications for Practical Applications: While the study advances scientific understanding, its implications for practical applications in environmental monitoring or other fields are not clearly discussed. Providing insights into how these findings could be applied in real-world scenarios would enhance the relevance of the research

  >>To be honest, it would be great to find a practical application for our approach to determine microbial death rates. Considering the cost aspect of the analyses I highly doubt that this approach will see an application in the real world soon.

- L304 – Love this statement. Though it does make sense that forest microbes would be more temperature sensitive than agricultural microbes who are frequently exposed to bare earth and high temperatures compared to forest microbes.

  >> It would make more sense that forest microbes are more temperature sensitive than agricultural microbes, however this is not reflected in the CUE. In Fig 3e, we do see a significant decrease in CUE in agricultural soils but not in forest soils.

---

## Author Response (AR3)

Dear editor and authors,

I have carefully read the revised manuscript (egusphere-2023-2302) and the author's responses to the editor and reviewers, and it is clear that this manuscript is worthy of publication in journal SOIL.

Thanks to the authors for improving measurements of microbial growth, death, and turnover by accounting for eDNA in soils. I believe that this new approach determining microbial death rates and dynamics of intracellular and eDNA will help to improve concepts and models of carbon dynamics in soils in the future.

In addition, in order for the reader to see the data clearly, I strongly recommend that the authors make the figures (1-3) in the manuscript clearer and more beautiful.

>> Thank you very much for your review and your suggestion to improve the figures. We have no removed the boxplots and dotpots from figure 2 and figure 3 and have also included a description of the X-axis labels (A20, F20, A30, ... ) in the figure caption. We have also increased the resolution of the figures.